# Incremental Diagnostic Value of CT Fractional Flow Reserve Using Subtraction Method in Patients with Severe Calcification: A Pilot Study

**DOI:** 10.3390/jcm10194398

**Published:** 2021-09-26

**Authors:** Yuki Kamo, Shinichiro Fujimoto, Yui O. Nozaki, Chihiro Aoshima, Yuko O. Kawaguchi, Tomotaka Dohi, Ayako Kudo, Daigo Takahashi, Kazuhisa Takamura, Makoto Hiki, Iwao Okai, Shinya Okazaki, Nobuo Tomizawa, Kanako K. Kumamaru, Shigeki Aoki, Tohru Minamino

**Affiliations:** 1Department of Cardiovascular Biology and Medicine, Juntendo University Graduate School of Medicine, Tokyo 113-8421, Japan; y-kawai@juntendo.ac.jp (Y.K.); y-nozaki@juntendo.ac.jp (Y.O.N.); caoshima@juntendo.ac.jp (C.A.); yukawagu@juntendo.ac.jp (Y.O.K.); tdohi@juntendo.ac.jp (T.D.); a.kudo.gt@juntendo.ac.jp (A.K.); d-takahashi@juntendo.ac.jp (D.T.); k-takamu@juntendo.ac.jp (K.T.); m-hiki@juntendo.ac.jp (M.H.); okaiwao@juntendo.ac.jp (I.O.); shinya@juntendo.ac.jp (S.O.); t.minamino@juntendo.ac.jp (T.M.); 2Department of Radiology, Juntendo University Graduate School of Medicine, Tokyo 113-8421, Japan; n-tomizawa@juntendo.ac.jp (N.T.); k-kumamaru@juntendo.ac.jp (K.K.K.); s-aoki@juntendo.ac.jp (S.A.); 3Japan Agency for Medical Research and Development-Core Research for Evolutionary Medical Science and Technology (AMED-CREST), Japan Agency for Medical Research and Development, Tokyo 100-0004, Japan

**Keywords:** coronary CT angiography, subtraction, fractional flow reserve, coronary artery disease, Agatston score

## Abstract

Although on-site workstation-based CT fractional flow reserve (CT-FFR) is an emerging method for assessing vessel-specific ischemia in coronary artery disease, severe calcification is a significant factor affecting CT-FFR’s diagnostic performance. The subtraction method significantly improves the diagnostic value with respect to anatomic stenosis for patients with severe calcification in coronary CT angiography (CCTA). We evaluated the diagnostic capability of CT-FFR using the subtraction method (subtraction CT-FFR) in patients with severe calcification. This study included 32 patients with 45 lesions with severe calcification (Agatston score >400) who underwent both CCTA and subtraction CCTA using 320-row area detector CT and also received invasive FFR within 90 days. The diagnostic capabilities of CT-FFR and subtraction CT-FFR were compared. The sensitivities, specificities, positive predictive values (PPVs), and negative predictive values (NPVs) of CT-FFR vs. subtraction CT-FFR for detecting hemodynamically significant stenosis, defined as FFR ≤ 0.8, were 84.6% vs. 92.3%, 59.4% vs. 75.0%, 45.8% vs. 60.0%, and 90.5% vs. 96.0%, respectively. The area under the curve for subtraction CT-FFR was significantly higher than for CT-FFR (0.84 vs. 0.70) (*p* = 0.04). The inter-observer and intra-observer variabilities of subtraction CT-FFR were 0.76 and 0.75, respectively. In patients with severe calcification, subtraction CT-FFR had an incremental diagnostic value over CT-FFR, increasing the specificity and PPV while maintaining the sensitivity and NPV with high reproducibility.

## 1. Introduction

Multiple methods for non-invasively calculating fractional flow reserve (FFR) have been developed based on coronary computed tomography angiography (CCTA) images, and all have been reported to add an incremental diagnostic value to conventional CCTA using invasive FFR as a reference [1,2,3,4,5,6]. However, variations have been reported in specificity and the positive predictive value, compared to sensitivity and the negative predictive value [7,8,9].

A FFR calculation algorithm was developed from CCTA acquired via 320-row area detector CT (320-ADCT) using fluid–structure interaction as a method for CT-derived FFR (CT-FFR). This is considered to be capable of setting conditions unique to each patient in CT-FFR calculations, based on the shape, movement, cross-sectional area, and changes in the volume of the coronary artery, by acquiring multiple optimum cardiac phases from 70–99% of the cardiac phase data within one heartbeat and analyzing these data based on the hierarchical Bayes and Markov chain Monte Carlo method [10,11]. In addition, on-site analysis at a workstation is possible by calculating the 1D computational fluid dynamics. The diagnostic performance of CT-FFR with the positivity criterion defined as the invasive FFR ≤0.8 has previously been demonstrated, with the rate of accurate diagnosis being significantly higher than that of conventional CCTA; however, similarly to other methods, the specificity was lower than the sensitivity [12,13]. Contributing factors may be overestimation of the severity of stenosis or underestimation of the vascular diameter due to the spatial resolution and the influence of artifacts generated by calcification in the case of CCTA [14]. We previously reported that the specificity of CT-FFR markedly decreases in cases with severe calcification (Agatston score ≥400) and the presence of calcified plaques was identified as the strongest factor predicting false positivity in CT-FFR [12,15].

A method termed “subtraction” has recently been developed in which the influence of calcification is removed from the vascular lumen in order to observe the degree of stenosis of lesions by differentiating non-contrast-enhanced CT information from contrast-enhanced CT information [16]. Improvements have been achieved in the diagnostic performance of CCTA for invasive coronary angiography by using the subtraction method in severe calcification cases [17]; however, it has not yet been applied to CT-FFR.

The present study investigated the incremental diagnostic value of CT-FFR evaluated via CCTA where calcification was removed using the subtraction method in patients with severe calcification (Agatston score ≥ 400).

## 2. Materials and Methods

### 2.1. Study Population

Data accounting for 70–99% of the R-R interval within one heartbeat, from which CT-FFR may be calculated, were collected from 1594 out of 2742 patients who were examined for suspected coronary artery disease by CCTA using 320-ADCT between 1 January 2016 and 31 December 2019. The coronary artery calcification score (Agatston score) measured using a non-contrast CT scan was 400 or higher in 264 patients. Following the exclusion of patients judged as having difficulty in breath-holding for 25 s before imaging and those with large variations in heart rate during breath-holding (judged as inappropriate for the subtraction method), the final number of patients from whom images were acquired using the subtraction method was 195.

Invasive FFR was performed within 90 days of CCTA in 42 out of the 195 patients, consent to participation in the study was obtained from 37 patients (53 vessels), and the CT-FFR analysis was ultimately performed on 32 patients (45 vessels).

The present study was approved by the institutional Human Research Ethics Committee and all participants gave written informed consent. All procedures followed the principles of the Declaration of Helsinki.

### 2.2. Subtraction CCTA Acquisition

Patients with a pre-scan heart rate of ≥60 beats per minute were orally administered 20 to 40 mg of metoprolol. If their heart rate remained ≥60 beats per minute after 1 h, they were given an intravenous injection of landiolol (0.125 mg/kg) (Corebeta; Ono Pharmaceutical, Tokyo, Japan). Patients for whom beta-blockers were contraindicated (due to severe aortic stenosis, systolic blood pressure < 90 mmHg, bronchial asthma, symptomatic heart failure, or advanced atrioventricular block) did not receive these treatments. All patients received 0.6 mg of nitroglycerin sublingually (Myocor spray; Toa Eiyo, Tokyo, Japan).

CCTA was performed using 320-row CT equipment (Aquilion ONE Vision Edition, or GENESIS Edition; Canon Medical Systems Corporation, Otawara, Japan) with a collimation of 320 × 0.5 mm. All scans were performed at the fastest gantry rotation time of 275 ms using the prospective ECG-gated axial scan mode.

Each patient underwent an unenhanced scan at a tube voltage of 120 kVp and a tube current of 250 mA for calcium scoring. Images were reconstructed with a slice thickness of 3.0 mm and increments of 3.0 mm.

Patients received 18.0 mg of iodine/kg/s of iopamidol (Iopamiron 370 mg of iodine/kg; Bayer Holding Ltd., Osaka, Japan). A contrast medium was injected for 12 s, followed by 30 mL of a saline chaser. Two CCTA scans were performed during the subtraction CCTA examination [16,18,19,20]. Patients were asked to hold their breath immediately after the contrast medium injection started. The first scan was performed 5 s after the contrast medium injection started. The bolus tracking method was used to select the scan timing for the second scan. The second scan was performed 2 s after the CT number for the descending aorta reached 270 Hounsfield units (HU). Patients were asked to continue holding their breath throughout the scan (≈25 s). The scanning parameters for CCTA were as follows: tube voltage, 100 kVp (body mass index <30 kg/m^2^) or 120 kVp (body mass index ≥ 30 kg/m^2^); target SD, 22.0; scan coverage 100–160 mm; acquisition window, 70–99% of the R-R interval. Half-reconstruction was performed with a slice thickness of 0.5 mm and an increment of 0.25 mm, using a medium-soft tissue kernel (FC04) with adaptive iterative dose reductions using three-dimensional processing (AIDR3D; Canon Medical Systems). In each scan, four phases (70, 80, 90, and 99%) were reconstructed for the CT-FFR analysis. In addition, the phase with the minimum number of artifacts was selected at the CT console using cardiac-phase search software (PhaseNavi; Canon Medical Systems Corporation) for the visual CCTA analysis.

The subtraction CCTA images were derived using dedicated software (^SURE^Subtraction; Canon Medical Systems). Specifically, volume datasets of all parts of the images obtained by pre-contrast CT and post-contrast CT were used to create the subtraction image by subtracting the CT value of each pixel in the pre-contrast CT image from the CT value of the corresponding pixel in the post-contrast CT image. Global non-rigid registration followed by local rigid registration was performed to obtain the subtraction image. As a result, the obtained subtraction images were images of the target segments with calcification only [19,20,21].

During processing, images were transferred to a workstation (Zio M900; Ziosoft Inc., Tokyo, Japan and Vitrea; Canon Medical Systems Corporation, Otawara, Japan). The mean effective dose was derived from the dose–length product multiplied by a conversion coefficient for the chest (κ = 0.014 mSv/mGy/cm) [22].

### 2.3. Calcium Scoring

A calcified lesion was defined as ≥3 contiguous pixels with a peak attenuation of at least 130 Hounsfield units (HU) [23]. Lesion scores from the left main, left anterior descending, left circumflex, and right coronary arteries were summed to obtain the total calcium score.

### 2.4. CCTA Interpretation

Cross-sectional and longitudinal curved multi-planar reformation images were both analyzed for plaque detection. Coronary artery segments with diameters of ≥2 mm were evaluated for the degree of stenosis. The percent degree of stenosis was assessed by obtaining the percent ratio of the stenotic lumen to the normal vessel diameter proximal or distal to the stenosis. Stenosis was measured at the angle showing the narrowest degree of stenosis. The degree of stenosis was evaluated by consensus by three experienced cardiologists who were unaware of the clinical data. Lesions with >50% stenosis were defined as significant. When a lesion stenosis was considered to be impossible to assess due to heavy calcification, it was classified as significant (>50% stenosis).

### 2.5. CT-FFR Analysis

CT-FFR was calculated using non-commercial software (CT-FFR; Canon Medical Systems). Using the phase with the minimum number of artifacts, the vascular central line and contours were automatically identified and manually corrected when necessary. Vessel segmentation was applied to the other three phases. The boundary condition was identified using variations in the vascular cross-sectional area in the images of the four different phases (70, 80, 90, and 99%). Pressure and flow values throughout the coronary artery were then calculated by performing a fluid analysis. CT-FFR was calculated for the original and the subtracted data. CT-FFR was calculated using a previously reported method [11,12,15,24].

The CT-FFR analyses were performed by observers who had more than 50 h of experience using this software. Observers were blinded to the invasive angiography and FFR findings.

### 2.6. Reproducibility Analysis

To evaluate inter- and intra-observer variabilities in the subtraction CT-FFR calculation, another operator who had more than 50 h of experience using this software performed post-processing for 30 consecutive vessels. The second operator also repeated post-processing for 30 consecutive vessels approximately 1 month after the first analysis, to evaluate intra-observer variability. In each case, for each vessel and for each operator, subtraction CT-FFR values were compared with those measured at the same position in invasive FFR. Anatomical landmarks, such as calcium deposits and/or side branches, were used to obtain subtraction CT-FFR at the same location for different operators.

### 2.7. Invasive FFR

Pressure measurements were performed using a 0.014-inch pressure guide wire (Verrata Pressure Guide Wire, Volcano Corp., San Diego, CA, USA) and suitable software (s5x™ Imaging System, Volcano Corp., San Diego, CA, USA). The pressure wire was calibrated and equalized with aortic pressure before being placed distal to the stenosis and in the distal third of the coronary artery being interrogated.

FFR was measured as the mean distal coronary pressure (Pd) divided by the mean aortic pressure (Pa) during maximal hyperemia. In brief, FFR was measured with a coronary pressure guide wire at maximal hyperemia induced by adenosine triphosphate (ATP) administered at 140 μg/kg/min for at least 2 min through a large forearm vein using an infusion pump until heart rate began to increase and the Pd/Pa ratio remained constant. Pressure wire pullback was performed to check for FFR at each lesion segment and pressure drift. If a Pd/Pa ratio <0.98 or >1.02 at the catheter tip was documented, the protocol mandated a repeat assessment. An FFR value of ≤0.8 was selected to define hemodynamically significant stenosis [25,26].

### 2.8. Definition of Risk Factors

Hypertension was defined as either systolic or diastolic blood pressure ≥ 140/90 mmHg or the use of antihypertensive medications. Diabetes mellitus was defined as fasting blood sugar ≥ 7.0 mmol/L (126 mg/dL), postprandial blood sugar ≥ 11.0 mmol/L (200 mg/dL), hemoglobin A1c ≥ 6.5% (47.5 mmol/mol), or the use of antidiabetic medications. Dyslipidemia was defined as total cholesterol ≥ 5.7 mmol/L (220 mg/dL), low-density lipoprotein cholesterol > 3.6 mmol/L (140 mg/dL), fasting triglycerides ≥ 1.7 mmol/L (150 mg/dL), high density cholesterol < 1.0 mmol/L (40 mg/dL), or the use of lipid-lowering medications. Smokers were defined as patients who had smoked during the past 1 year from the time of CCTA acquisition.

### 2.9. Statistical Analysis

Continuous data were expressed as the mean ± standard deviation (SD). If the variables were non-normally distributed, the median and quartile values were used. When the median and quartile data were 0, the maximum and minimum results were added in the form of the median (quartile; range). Categorical data were expressed as frequencies (percentages). Intraclass correlation coefficients were used to evaluate inter- and intra-observer variabilities for the subtraction CT-FFR analysis. The sensitivities, specificities, positive predictive values, negative predictive values, and diagnostic accuracy values of CCTA > 50% vs. subtraction CCTA > 50% vs. CT-FFR vs. subtraction CT-FFR ≤ 0.8, with respect to detecting hemodynamically significant stenosis defined as invasive FFR ≤ 0.8, were calculated. Diagnostic accuracy values using the area under the curve (AUC) of the receiver operating characteristic (ROC) curve to detect invasive FFR ≤ 0.8 were compared for CCTA > 50% vs. subtraction CCTA > 50% vs. CT-FFR ≤ 0.8 vs. subtraction CT-FFR ≤ 0.8 using the DeLong test, and *p*-values of <0.05 were considered to be significant. The statistical analyses were performed using JMP software for Windows (SAS Institute Inc., Cary, CA, USA).

## 3. Results

### 3.1. Patient and Scan Characteristics

The patient and scan characteristics are shown in Table 1. The mean age of patients was 70.8 ± 7.8 years and the mean Agatston score was 1014.6 (523.9–1382.5). Twenty-two patients (68.8%) had taken β blockers before the acquisition of images and the mean heart rate at acquisition was 54.0 ± 4.6. None of the patients were administered intravenous iopamidol before imaging. All patients received nitroglycerin sublingually before imaging. The mean radiation exposure dose was 4.2 ± 1.1 mSv.

### 3.2. Vessel Characteristics

Patient-based analysis gave the following results: CCTA > 50% (31 patients (96.9%)), subtraction CCTA > 50% (22 (68.8%)), CT-FFR ≤ 0.8 (19 (59.4%)), and subtraction CT-FFR ≤ 0.8 (12 (37.5%)). Eleven patients (34.4%) showed invasive FFR ≤ 0.8.

Vessel-based analysis gave the following results: CCTA > 50% (42 vessels (93.3%)), subtraction CCTA > 50% (32 (71.1%)), CT-FFR ≤ 0.8 (25 (55.6%)), and subtraction CT-FFR ≤ 0.8 (20 (44.4%)). Thirteen vessels (28.9%) showed invasive FFR ≤ 0.8 (Table 2).

### 3.3. Diagnostic Accuracy of CCTA Findings, CT-FFR, and Subtraction CT-FFR

Table 3 shows the measurements of the diagnostic performances of CCTA > 50%, subtraction CCTA > 50%, CT-FFR ≤ 0.8, and subtraction CT-FFR ≤ 0.8 in detecting hemodynamically significant stenosis defined as invasive FFR ≤ 0.80.

In the patient-based analysis (Table 3a), the sensitivities, specificities, PPV, NPV, and accuracy values of CCTA > 50%, subtraction CCTA > 50%, CT-FFR ≤ 0.8, and subtraction CT-FFR ≤ 0.8 were 100% vs. 83.3% vs. 90.9% vs. 90.9%, 10.0% vs. 30.0% vs. 57.1% vs. 90.5%, 40.0% vs. 41.7% vs. 52.6% vs. 83.3%, 100% vs. 75.0% vs. 92.3% vs. 95.0%, and 43.8% vs. 50.0% vs. 68.8% vs. 90.6%, respectively.

In the vessel-based analysis (Table 3b), the sensitivities of CCTA > 50%, subtraction CCTA > 50%, CT-FFR ≤ 0.8, and subtraction CT-FFR ≤ 0.8 were 100% vs. 84.6% vs. 84.6% vs. 92.3%, the specificities were 9.4% vs. 31.3% vs. 59.4% vs. 75.0%, the PPV scores were 31.0% vs. 33.3% vs. 45.8% vs. 60.0%, the NPV scores were 100% vs. 83.3% vs. 90.5% vs. 96.0%, and the accuracy values were 35.6% vs. 46.7% vs. 66.7% vs. 80.0%, respectively.

Figure 1 shows that the vessel-based AUCs for CCTA > 50%, subtraction CCTA > 50%, CT-FFR ≤ 0.8, and subtraction CT-FFR ≤ 0.8 for invasive FFR ≤ 0.8 were 0.55 (95% confidence interval (CI): 0.50–0.60) vs. 0.60 (95% CI: 0.46–0.73) vs. 0.70 (95% CI: 0.57–0.84) vs. 0.84 (95% CI 0.73–0.94), respectively. Significant differences were noted between CCTA > 50% vs. CT-FFR (*p* = 0.02), CCTA > 50% vs. subtraction CT-FFR ≤ 0.8 (*p* < 0.01), CT-FFR ≤ 0.8 vs. subtraction CT-FFR ≤ 0.8 (*p* = 0.04), and subtraction CCTA > 50% vs. subtraction CT-FFR ≤ 0.8 (*p* < 0.01).

A representative case is shown in Figure 2.

### 3.4. Inter-Observer and Intra-Observer Reproducibility

In the analysis of 30 consecutive vessels, the correlation coefficient of inter-intra observer evaluation was 0.76 and the intra-observer–intraclass correlation coefficient was 0.75.

## 4. Discussion

To the best of our knowledge, this is the first study to apply the subtraction method to CT-FFR. Since the specificity of CT-FFR has previously been reported to be lower than the sensitivity using invasive FFR as a reference [12,13], unnecessary revascularization may result in an increase in false positive cases only, based on the results of CT-FFR. To overcome this problem, we reported the influence of pre-test probability on diagnostic performance as well as improvements in diagnostic performance using the correction formula for CT-FFR [24], and we also demonstrated that the strongest factor associated with false positivity was the presence of calcification [15]. Thus, we hypothesized that false positivity may be reduced by analyzing CT-FFR in images from which coronary arterial calcification had been removed using the subtraction method, particularly in cases with severe calcification. The subtraction CT-FFR method achieved a higher specificity and PPV than CT-FFR analyzed using conventional CCTA images, while maintaining the sensitivity and NPV, thereby reducing the false positive cases from nine to two patients in the patient-based analysis and from thirteen to eight lesions in the vessel-based analysis. Therefore, subtraction CT-FFR significantly increased the diagnostic accuracy, suggesting that overestimations of the degree of stenosis and underestimations of the vascular diameter due to the influence of spatial resolution and artifacts generated by calcification in the CT-FFR analysis are the major factors leading to false positive cases, particularly in cases with severe calcification. However, in a previous study using FFR_CT_ (HeartFlow Inc., Redwood City, CA, USA), no significant difference in diagnostic performance due to the severity of calcification was noted, while the diagnostic performance of FFR_CT_ tended to be lower when limited to the subgroup with severe calcification similar to the calcification in this study [27]. The CT-FFR technique used in the present study is an on-site local computational analysis technique and the contours of the vascular wall and inner lumen are analyzed semi-automatically; therefore, manual correction may be necessary depending on individual cases. The images with severe calcification required more manual correction in the present study. The objectivity and accuracy of not only automatic extraction but also manual correction can be improved in subtraction images. We previously reported that analytical accuracy is stabilized by the training of analysts for CT-FFR [28,29] and that the inter-observer and intra-observer reproducibility of subtraction CT-FFR was also favorable.

However, among the 53 vessels remaining after participant consents were obtained, 3 vessels for both CT-FFR and subtraction CT-FFR, 1 vessel for CT-FFR alone, and 4 vessels for subtraction CT-FFR alone could not be analyzed. One of the reasons was that in conventional CT-FFR, the boundary between the inner lumen and wall became unclear due to calcification-induced artifacts and the inner lumen was visualized as narrower than it actually was. Moreover, in subtraction CT-FFR, calcified lesions were visualized as larger, due to the misregistration caused by the blurring of images in the differentiation of non-contrast-enhanced CT images from contrast-enhanced CT images in which the inner lumen is visualized as narrower. This may also be a factor contributing to false positivity in the 8 out of 45 vessels from which the analytical results of subtraction CT-FFR were acquired. Since non-contrast-enhanced and contrast-enhanced CT images cannot be simultaneously acquired, misregistration may be due to factors such as the heart rate [30], poor breath-holding [31], and body movement during imaging.

Misregistration is an important issue in the use of the subtraction method. In a previous study, misregistration was noted in approximately 50% of the segments of CCTA images acquired using the subtraction method and the frequency of misregistration increased as the lesion became a distal site [30]. However, in the present study, misregistration was found in only approximately 15% of vessels. To reduce the misregistration and increase the diagnostic accuracy of subtraction CT-FFR, appropriate cases should be selected.

Moreover, the radiation exposure dose was higher in the subtraction method than in conventional imaging because images were acquired twice for comparisons between contrast-enhanced and non-contrast-enhanced imaging. A previous study reported that the effective radiation dose in subtraction CCTA acquired using the single breath-holding method was 5.2–10 mSv [16]; however, the effective radiation dose was reduced to 4.2 ± 1.1 mSv in the present study by applying low-voltage imaging at 100 kVp in patients with a body mass index of 30 or lower [17], and this method was considered to be acceptable for clinical use.

### Limitations

There are some limitations that need to be addressed. This was a single-center study with a small number of subjects. Among patients with severe calcification, CT-FFR analysis was only performed on the images with an R-R interval of 70–99% in the diastolic phase of one heartbeat. Furthermore, acquisition using the subtraction method was limited to those patients who were judged to be capable of holding their breath for at least 25 s. Accordingly, 195 out of 264 patients with severe calcification could actually be imaged using the subtraction method. In addition, although the radiation dose was relatively low because in most of the patients CCTA was performed with a tube voltage of 100 kVp as previously described, a higher radiation dose than that for ordinary CCTA is one of the weak points of this subtraction method. This method was only analyzed using 320-row CT equipment and the specific software mentioned, which is likely to represent limited versatility. Furthermore, the indication of invasive coronary angiography and invasive FFR within 90 days depended on the judgment of the attending physicians according to the results of CCTA, suggesting that case selection was biased.

## 5. Conclusions

By analyzing CT-FFR images of severely calcified lesions (Agatston score ≥400) acquired using the subtraction method, the number of false positive CT-FFR cases was reduced and the diagnostic performance was also significantly improved.

## Figures and Tables

**Figure 1 jcm-10-04398-f001:**
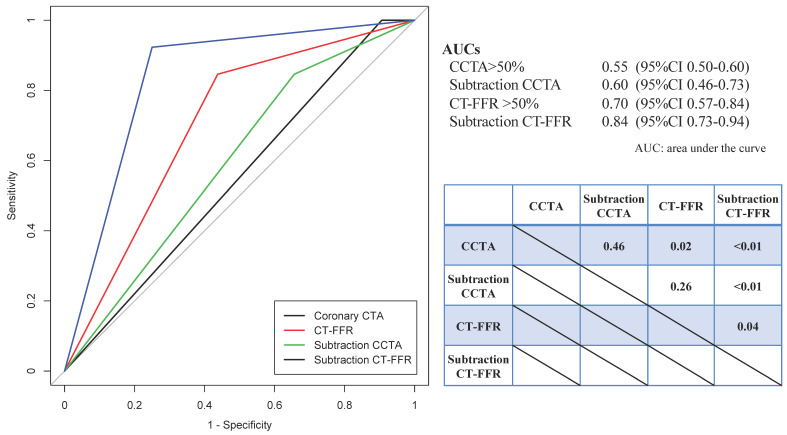
Comparison of areas under the curve (AUC) for the receiver operating characteristic curves of CCTA > 50%, subtraction CCTA > 50%, CT-FFR ≤ 0.8, subtraction CT-FFR ≤ 0.8.

**Figure 2 jcm-10-04398-f002:**
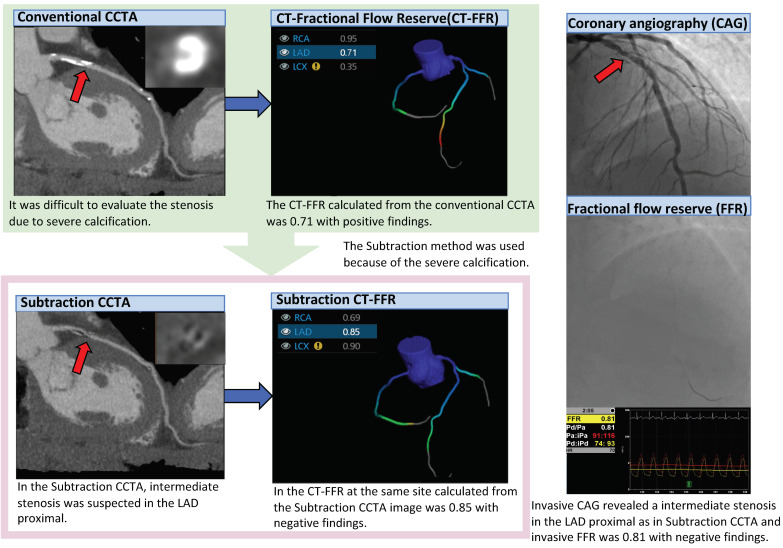
Representative case of subtraction CT-FFR. Since the Agatston score was 738.8, CCTA was performed using the subtraction method. In conventional CCTA, a calcified plaque was found in the LAD proximal.

**Table 1 jcm-10-04398-t001:** Patient and scan characteristics.

32 Patients	
Age (years)	70.8 ± 7.8
Gender (M/F)	22/11
Body mass index (Kg/m^2^)	24.3 ± 3.1
Diabetes mellitus (%)	16 (50.0)
Hypertension (%)	22 (68.8)
Dyslipidemia (%)	21 (65.6)
Smoking	
current/former/never	2/17/13
Heart rate (bpm)	54.0 ± 4.6
Total CACS ^1^ (Agatston score)	1014.6 (523.9–1382.5)
β blocker administered (%)	
None	10 (31.3)
Oral	22 (68.8)
Intravenous	0 (0)
Nitrates administered	32 (100)
Tube voltage (%)	
100 kVp	27 (84.4)
120 kVp	5 (15.6)
Tube current (mA)	559.6 ± 43.8
DLPe ^2^ (mGy.cm)	299.3 ± 80.3
Effective dose (mSV)	4.2 ± 1.1

^1^ CACS: coronary artery calcium score; ^2^ DLPe: extended dose–length product.

**Table 2 jcm-10-04398-t002:** Vessel characteristics.

32 Patients, 45 Vessels		
	Patient	Vessel
CCTA ^1^ maximum stenosis > 50% (%)	31 (96.9)	42 (93.3)
Subtraction CCTA maximum stenosis > 50% (%)	22 (68.8)	32 (71.1)
CT-FFR ^2^ ≤ 0.8 (%)	19 (59.4)	25 (55.6)
Subtraction CT-FFR ≤ 0.8 (%)	12 (37.5)	20 (44.4)
Invasive FFR ≤ 0.8 (%)	11 (34.4)	13 (28.9)
RCA/LAD/LCX	13/20/12	
CACS ^3^		
RCA ^4^	343.7 (124.0–632.3)	
LAD ^5^	348.4 (243.0–611.0)	
LCX ^6^	116.5 (55.1–252.3)	

^1^ CCTA: coronary computed tomography angiography; ^2^ FFR: fractional flow reserve; ^3^ CACS: coronary artery calcium score; ^4^ RCA: right coronary artery; ^5^ LAD: left anterior descending artery; ^6^ LCX: left circumflex artery.

**Table 3 jcm-10-04398-t003:** Diagnostic accuracies of CCTA findings, CT-FFR, subtraction CCTA and subtraction CT-FFR on a per patient and per vessel basis.

(a) Per Patient				
	CCTA ^1^ findings	Subtraction CTA	CT-FFR ^2^	Subtraction CT-FFR
True positive (*n*)	12	10	10	10
True negative (*n*)	2	6	12	9
False positive (*n*)	18	14	9	2
False negative (*n*)	0	2	1	1
Sensitivity (%)	100	83.3	90.9	90.9
True negative (%)	10.0	30.0	57.1	90.5
False positive (%)	40.0	41.7	52.6	83.3
False negative (%)	100	75.0	92.3	95.0
Accuracy (%)	43.8	50.0	68.8	90.6
**(b) Per Vessel**				
	CCTA findings	Subtraction CTA	CT-FFR	Subtraction CT-FFR
True positive (*n*)	13	11	11	12
True negative (*n*)	3	10	19	24
False positive (*n*)	29	22	13	8
False negative (*n*)	0	2	2	1
Sensitivity (%)	100	84.6	94.6	92.3
True negative (%)	9.4	31.3	59.4	75.0
False positive (%)	31.0	33.3	45.8	60.0
False negative (%)	100	83.3	90.5	96.0
Accuracy (%)	35.6	46.7	66.7	80.0

^1^ CCTA: coronary computed tomography angiography; ^2^ FFR: fractional flow reserve.

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
