# Peer review of "Incremental Diagnostic Value of CT Fractional Flow Reserve Using Subtraction Method in Patients with Severe Calcification: A Pilot Study"

_jcm, 2021, doi:10.3390/jcm10194398_

Round 1

Reviewer 1 Report

A study assessing diagnostic performance of Subtraction CT-FFR compared to traditional angio-FFR that enrolled 32 patients with 45 severely calcified coronary lesions.

The study is comprehensively presented, Results are adequately discussed.

However, the concept of Subtraction method shuld be desribed in a more detailed way to let the readers know what is actually being 'subtraced': are it pixels/voxels of certain attenuation range?

Secondly, the study population is small and it should be considered a pilot study. I would suggest inluding this information in the title.

Also, a piece of information should be included whether CCTA data was blinded when angio-FFR was performed.

In Discussion (lines 276-278), the Authors cite the 2015 study by Norgaard et al, showing that no significant difference due to severity of calcification was noted in diagnostic performance of CT-FFR. It should be underlined, however, that Calcium Score in the study by Norgaard et al. was much lower than in this study - only the 4th Quartile (416-3599 Ag) is a subgroup corresponding with the study population (523-1382 Ag), and the diagnostic performance in the 4th quartile was lower. Norgaard et al. themselves state that 'there was a tendency towards declining diagnostic accuracy and specificity with increasing AS levels'. These findings should be included in the discussion.

Limitations should be presented in a separate paragraph, since there are many. Please include higher radiation dosis in the limitations and the need of specific software required to use this method.

Minor considerations:

  • please add 'previously' in the line 263 to indicate that you refer to your previous research;
  • please remove 'with unknown coronary artery disease' form line 71-72 as non-contributory;
  • Fig. 2, left panels include a cross-section vew, however, it is not marked which part of the lesion/vessel they refer to.

Author Response

Reviewer 1

1. The concept of Subtraction method should be described in a more detailed way to let the readers know what is actually being 'sub-traced': are it pixels/voxels of certain attenuation range?

1. Response to Reviewer 1

Thank you for your comment. We added the detailed information of sub-traced method in the method paragraph.

2. The study population is small and it should be considered a pilot study. I would suggest including this information in the title.

2. Response to Reviewer 1

Thank you for your comment. We added ‘a pilot study’ on the title.

3. A piece of information should be included whether CCTA data was blinded when angio-FFR was performed.

3. Response to Reviewer 1

Thank you for your comment. But we have already indicated this information in the method paragraph. (line 149-151)

4. Calcium Score in the study by Norgaard et al. was much lower than in this study - only the 4th Quartile (416-3599 Ag) is a subgroup corresponding with the study population (523-1382 Ag), and the diagnostic performance in the 4th quartile was lower. Norgaard et al. themselves state that 'there was a tendency towards declining diagnostic accuracy and specificity with increasing AS levels'. These findings should be included in the discussion.

4. Response to Reviewer 1

Thank you for your comment. As you mentioned, the AS in the 4th quartile subgroup was similar but even lower than out study. We added this discussion.

5. Limitations should be presented in a separate paragraph, since there are many. Please include higher radiation doses in the limitations and the need of specific software required to use this method.

5. Response to Reviewer 1

Thank you for your comment. We separated the limitation paragraph and added them, too.

6. Please add 'previously' in the line 263 to indicate that you refer to your previous research.

6. Response to Reviewer 1

Thank you for your comment. We revised it.

7. Please remove 'with unknown coronary artery disease' form line 71-72 as non-contributory.

7. Response to Reviewer 1

Thank you for your comment. We revised it.

8. Fig. 2, left panels include a cross-section view, however, it is not marked which part of the lesion/vessel they refer to.

8. Response to Reviewer 1

Thank you for your comment. We added the red arrow on the left panels.

Reviewer 2 Report

The paper tat was submitted to our evaluation is a very interesting one, covering the field of non-invasive determination of FFR based on coronary computed tomography angiography. As the authors have underlined, the subject is one of great novelty, supporting the use of the Subtraction method to CT-FFR. I believe that the study is well presented, with accurate statistical data and results. Therefore, I recommend acceptance of the paper in present form. 

Author Response

Reviewer 2

Thank you for your kind comment. We would like to keep on trying this study, the subtraction method.
